# Maternal body mass index and necrotizing enterocolitis: A case-control study

Katherine Stumpf[1]*, Priya Sharma[1], L. Steven Brown[2], Luc P. Brion[1], Julie Mirpuri[1]

**1** Department of Pediatrics, Division of Neonatal-Perinatal Medicine, University of Texas Southwestern Medical Center, Dallas, Texas, United States of America, **2** Parkland Health and Hospital System, Dallas, Texas, United States of America

* katherine.stumpf@utsouthwestern.edu

## Abstract

### Introduction

Our aim was to determine if maternal body mass index (BMI) is associated with necrotizing enterocolitis (NEC) in a large urban delivery center.

### Methods

This single center retrospective case-control study included 291 infants under gestational age of 33 weeks admitted to the neonatal intensive care unit (NICU) during a 10-year period. Cases of stage 2 and 3 NEC were matched at a ratio of 2 controls (n = 194) to 1 case (n = 97). Maternal BMI was categorized as normal ($\leq$24.9), overweight (25–29.9) and obese ($\geq$30). Chi-square and stepwise logistic regression were used for analysis. A power analysis was performed to determine if sample size was sufficient to detect an association.

### Results

Stepwise logistic regression demonstrated no association between NEC and maternal obesity. Maternal hypertension, pre-eclampsia, premature rupture of membranes, maternal exposure to antibiotics, placental abruption and gestational diabetes were not associated with NEC. Power analysis showed the sample size was sufficient to detect an association of NEC with maternal BMI in three groups analyzed. In this case-control study, there was an association between NEC and maternal overweight but not obesity at delivery.

### Discussion

Our results did not show a significant association of NEC with maternal obesity. The percent of overweight and obese mothers prior to pregnancy and at delivery was significantly higher in our population than the national average and may be responsible for the limited ability to reveal any association between maternal obesity and NEC.

**Data Availability Statement:** De-identified data set has been uploaded as "supporting information". Data set is also available upon request. Point of contact for data inquiries (non-author):

HRPP@utsouthwestern.edu (This email contacts human subjects research protection program).

**Funding:** This study was supported in part by NIH NIDDK R01 DK121975 to J.M. The funder had no role in study design, data collection and analysis, decision to publish, or preparation of the manuscript.

**Competing interests:** The authors have declared that no competing interests exist.

## Introduction

Over 40% of women in the United States are now obese [1, 2] and over 50% of women of child-bearing age are either overweight or obese. Maternal obesity has been linked to increased risk of spontaneous and medically indicated preterm delivery [1], birth asphyxia [2], delivery room resuscitation [3], large for gestational age infants, congenital anomalies, admission to the neonatal intensive care unit (NICU), and perinatal death [4–7]. Furthermore, even women who are overweight (body mass index [BMI] 25–29.9) have an increased risk for preterm delivery [8]. A recent study also showed that maternal overweight was associated with a higher odds of lower language scores [9].

Maternal obesity has been associated with increased risk of neonatal necrotizing enterocolitis (NEC), however the literature thus far has been mixed [10, 11]. Maternal obesity is linked to increased IL-6, TNF-α, and leptin. Maternal increase in BMI is also associated with increased IL-1β and contributes to a heightened inflammatory state. Downstream effects include decreased placental VEGF signaling, reduced angiogenesis, placental hypoperfusion-insufficiency, and a hypoxic state. This inflammatory environment combined with placental hypoperfusion is thought to be the pathophysiological basis of increased neonatal morbidity in infants born to obese mothers [12]. Given infants with NEC have an estimated mortality rate of 20 to 30% [13, 14], identifying risk factors for the disease is beneficial for clinicians to stratify risk of infants in the NICU. Further, high maternal BMI (overweight and obesity) is a risk factor amenable to public health initiatives and intervention.

The aim of this case-control study was to determine if maternal BMI is associated with neonatal NEC in a large urban delivery hospital. We hypothesized that infants born to obese mothers would have increased NEC when compared to infants born to normal weight mothers.

## Materials and methods

This was a single-center, retrospective case-control study. Following approval from the institutional review board at the University of Texas Southwestern Medical School and Parkland Health and Hospital System, data were collected from a database established in 1980 and maintained/validated by the Division of Neonatal-Perinatal Medicine in the Department of Pediatrics as well as review of the electronic health record (EHR) [15]. We identified all infants less than 33 weeks gestational age admitted to Parkland Hospital NICU over a ten-year period between January 1, 2009, to December 31, 2018, with a diagnosis of NEC at any time during their admission. All NEC cases were NEC stage 2 and 3 by the modified Bell criteria [16]; thus, all infants identified as cases had documented systemic, gastrointestinal and radiologic findings consistent with NEC. Controls were infants admitted to the NICU from the same time period who did not have a gastrointestinal diagnosis, and were matched by gender, gestational age, weight and year of birth; the case to control ratio was 1 to 2.

Medical records for infants and mothers were reviewed for neonatal and maternal demographic data, maternal BMI and maternal risk factors. Maternal risk factors included maternal hypertension, pre-eclampsia, premature rupture of membranes, maternal exposure to antibiotics, placental abruption and gestational diabetes. Maternal BMI prior to pregnancy, during each prenatal visit and on day of delivery was collected, if available. BMI was calculated as below:

$$BMI = Weight(kg) \div \lceil Height \rceil (m^2)$$

Maternal BMI on day of delivery was documented for 96 cases and all 226 controls; maternal BMI at a prenatal visit was documented for 93 cases and all 226 controls; maternal BMI prior to

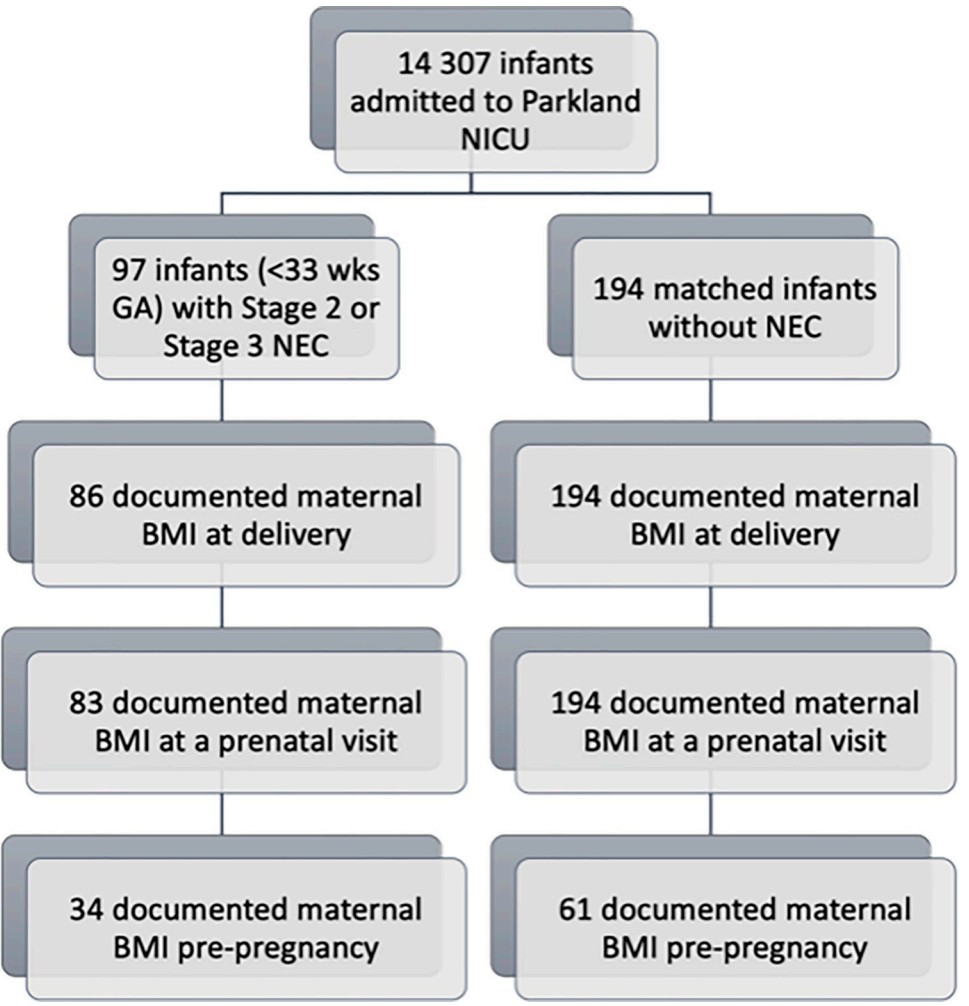

**Fig 1. Documented maternal BMI: Pre-pregnancy, at prenatal visits and at delivery.**

pregnancy was documented for 38 cases and 74 controls (Fig 1). Per the World Health Organization designations, maternal BMI was categorized into normal (BMI ≤24.9), overweight (BMI 25–29.9) and obese (BMI ≥30). The rate at which maternal BMI increased during pregnancy was calculated for mothers with both a pre-pregnancy and delivery BMI documented.

Statistical analysis included chi-square test for univariate analysis of categorical variables and t-test for continuous variables. Stepwise logistic regression analysis was used to determine the adjusted odds ratio (aOR) and the 95% confidence interval (CI) of NEC for the significant variables in the univariate analysis. Statistical analysis was done using SPSS version 25 (IBM, Inc., Armonk, NY) with two-tailed tests and $p<0.05$ considered statistically significant. Power analysis using SAS power procedure (SAS 9.4, SAS Institute, Cary, NC) was used to determine if the available sample size was sufficient to detect an association of NEC with maternal BMI classified in three groups.

## Results

97 infants diagnosed with NEC were identified over the 10-year period and were matched with 194 control infants without a gastrointestinal diagnosis. A total of 291 infant charts, in addition

**Table 1. Maternal (A) and Infant (B) characteristics of patients with NEC compared to matched controls.**

| | NEC (N = 97), n (%) | Controls (N = 194), n (%) | Chi-square p-value |
|---|---|---|---|
| **(A) Maternal Characteristics** | | | |
| Hypertension | 22 (23) | 56 (29) | 0.26 |
| Preeclampsia | 38 (39) | 76 (39) | 1.00 |
| Progesterone | 1 (1) | 3 (2) | 1.00 |
| Tocolytics | 1 (1) | 1 (1) | 1.00 |
| Magnesium | 46 (47) | 103 (53) | 0.36 |
| PROM | 15 (16) | 42 (22) | 0.21 |
| Antibiotics | 54 (56) | 107 (55) | 0.93 |
| Placental Abruption | 7 (7) | 22 (11) | 0.27 |
| Gestational Diabetes | 7 (7) | 24 (12) | 0.18 |
| Urine/Meconium Tox positive | 4 (4) | 4 (2) | 0.45 |
| BMI at delivery | | | 0.01** |
| ≤24.9 | 6 (7) | 30 (16) | |
| 25–29.9 | 37 (43) | 52 (27) | |
| ≥30 | 43 (50) | 112 (58) | |
| **(B) Infant Characteristics** | | | |
| **Birth weight (g)** | | | 1.00 |
| <1000 | 43 (44) | 86 (44) | |
| 1000–1399 | 38 (39) | 76 (39) | |
| 1400–1799 | 12 (12) | 24 (12) | |
| 1800–2199 | 3 (3) | 6 (3) | |
| 2200–2599 | 1 (1) | 2 (1) | |
| **Sex** | | | 1.00 |
| Female | 44 (45) | 88 (45) | |
| Male | 53 (55) | 106 (55) | |
| **Gestational age at birth (weeks)** | | | 1.00 |
| 24–26 | 28 (29) | 56 (29) | |
| 27–29 | 40 (41) | 80 (41) | |
| 30–32 | 29 (30) | 58 (30) | |
| **Small for gestational age** | 16 (17) | 20 (10) | 0.13 |
| **Intrauterine growth retardation** | 12 (12) | 21 (11) | 0.70 |
| **Multiples** | 20 (21) | 39 (20) | 0.92 |
| **Race/ethnicity** | | | 0.16 |
| Non-Hispanic White | 6 (6) | 10 (5) | |
| Hispanic White | 58 (60) | 136 (70) | |
| Non-Hispanic Black | 29 (30) | 46 (24) | |
| Other | 4 (4) | 2 (1) | |
| **Hispanic** | 58 (60) | 136 (70) | 0.08 |
| **Antenatal Steroids** | 61 (63) | 103 (53) | 0.11 |

BMI: body mass index; Tox: toxicology; PROM: premature rupture of membranes

to the 291 associated maternal charts, were reviewed. Table 1 shows the demographic information for the case and control infants. In addition to matched characteristics (birth weight, gestational age, gender and year of birth), there was no statistically significant difference amongst both groups for race, ethnicity, SGA status, intrauterine growth restriction (IUGR) status, or the use of antenatal steroids.

**Table 2. Adjusted odds ratio (aOR) of NEC with logistic regression analysis of maternal BMI at delivery.**

| Maternal BMI | aOR (95% CI) | P-value |
|---|---|---|
| **Primary outcome** | | 0.01 |
| Normal | reference | - |
| Overweight | 3.6 (1.3–9.4) | 0.01** |
| Obese | 1.9 (0.8–4.9) | 0.18 |
| **Sensitivity analysis** | | |
| Normal | reference | - |
| Overweight combined with obese | 2.4 (1.0–6.1) | 0.06 |

The multivariate analysis used the above variables to determine any significant association with infants who develop NEC. Using a p-value of <0.05, mothers who were overweight, compared to mothers with normal BMIs, had an increased risk of developing NEC in the multivariate analysis.

## Maternal factors and association with NEC

In the univariate analysis, maternal hypertension, pre-eclampsia, premature rupture of membranes, maternal exposure to antibiotics, placental abruption, positive toxicology (either urine or meconium) and gestational diabetes were not associated with increased odds for NEC (Table 1). Maternal BMI at delivery was associated with NEC (p-value = 0.01). Pairwise comparison with Bonferri correction demonstrated that maternal overweight was specifically associated with NEC.

In the logistic regression model (Table 2), overweight mothers had a statistically significant ($p = 0.01$) higher odds of NEC (aOR = 3.6, CI 1.3, 9.4) than normal BMI mothers. This remained significant even after further correcting for SGA, race and presence of hypertension. Obese mothers had an aOR = 1.9, CI 0.8, 4.9, which was not statistically significant (p = 0.18). Sensitivity analysis grouping overweight and obese and comparing to normal BMI had an aOR = 2.4, CI 1.0, 6.1, which was not statistically significant ($p = 0.06$). Power analysis showed that the available sample size was sufficient to detect an association of NEC with maternal BMI classified in 3 groups using logistic regression with aOR of 1.9, two-tailed test, $p<0.05$ and power of 0.92, despite observed skewed distribution with only 12% of mothers with normal BMI at delivery.

Calculated change in BMI per week throughout pregnancy showed no statistically significant difference between both groups (Table 3). Of note, while most mothers had documented BMI at delivery, fewer mothers had documented BMI at prenatal visits and prior to pregnancy; only 95 (of total 291) pre-pregnancy BMIs were available for analysis (Fig 1). At delivery, only 7% of case mothers and 16% of control mothers had normal BMI while all other mothers were overweight or obese (Fig 2).

## Discussion

The pathophysiology of NEC is multifactorial, involving a genetic predisposition, immaturity, inflammation and microbial colonization [13]. Our hypothesis for this study was that maternal

**Table 3. Change in maternal BMI during pregnancy for cases and controls.**

| | Infants with NEC (N = 78) | Controls (N = 180) | T-test p-value |
|---|---|---|---|
| Change in maternal BMI per week (Mean ± SD) | 0.24 ± 0.31 | 0.20 ± 0.16 | 0.28 |

The average change in maternal BMI per week during pregnancy was calculated from the pre-pregnancy or prenatal BMI and delivery BMI. Among both the case and control groups, there was not statistically significant difference in the change in maternal BMI through pregnancy.

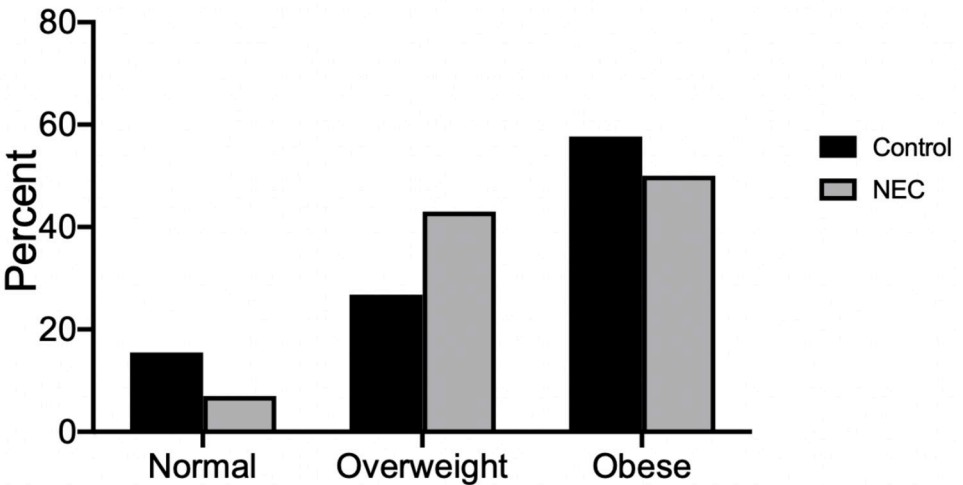

**Fig 2. Percent of mothers who have normal, overweight or obese BMI at delivery.**

obesity, which causes systemic inflammation and is correlated with changes in gut flora [17], puts infants at higher risk of developing NEC, possibly by causing systemic inflammation in the infant or changing the infant's microbiome. Another pathophysiological basis for the increased risk of NEC in offspring of obese or overweight mothers is reduction of fetal blood flow from a compromised placenta resulting in hypoperfusion to the fetal intestine in a subset of obese mothers [12]. Our results did not show a significant risk of developing NEC in infants with obese mothers The percent of overweight and obese mothers prior to pregnancy and at delivery is significantly higher in our population than the national average [18] and may be responsible for the limited ability to reveal any association between maternal obesity and NEC.

One surprising finding from this study was the association of NEC with overweight but not with obesity, in both simple and multiple logistic regression. Sample size was sufficient to detect an increased odds of NEC with BMI with an aOR of 1.9 despite observed skewed distribution. Since SGA and race have been shown to be associated with the development of NEC, we further corrected for this in our model but still found significance. In a recently published multi-center cohort study [19] of 22 to 28-week infants born in 2016–2018, Chawla et al showed that pre-pregnancy obesity was associated with decreased adjusted odds of infant survival and decreased odds of survival without NEC in comparison with overweight or normal BMI. Lack of association in models that included GA and SGA suggests that the effect of increased BMI on survival and survival without NEC was in part mediated by GA and size for GA. Lack of a significant association of maternal obesity with NEC in our study (in which we matched controls and cases for GA and birthweight) is consistent with mediation of the effect of maternal BMI on NEC by GA and size for age. Further studies dissecting the role of pre-pregnancy BMI and outcomes are needed.

It is also possible women who were overweight at delivery gained weight at a faster rate than their obese or normal weight counterparts. Based on the most recent American College of Obstetricians and Gynecologists (ACOG) recommendations, normal weight women should gain 25 to 35 pounds through the pregnancy, overweight women should gain 15 to 25 pounds and obese women should gain 11 to 20 pounds [20]. To determine if the rate of BMI rise plays a part in developing NEC, we evaluated this trend during pregnancy by comparing the pre-pregnancy weight or first prenatal visit weight with the maternal weight at delivery. While we

did not find a statistically significant difference between both groups' rates of BMI rise, our sample for this part of the study was smaller (78 cases, 180 controls). Parkland Hospital converted to electronic medical records at the beginning of our data collection period, thus few mothers had documented pre-pregnancy BMI during or prior to 2009. In addition, all pre-pregnancy BMI were collected from previous visits to our hospital system and many mothers did not establish care in our system until after they were pregnant. Another limitation of our study is the larger proportion of Hispanic mothers in our population. The disproportionate representation of a specific ethnic group within our sample may limit the broader generalizability of our results.

## Conclusion

In conclusion, we found an association of NEC with overweight but not with obesity. The unique case-control study design controlling for GA and birthweight and unique population with a very small proportion of women with normal BMI may have masked an association of NEC with obesity. This study contributes significantly to the expanding body of evidence emphasizing the importance of maternal BMI and health in determining infant outcomes. Prospective studies including pre-pregnancy BMI and the rate of BMI change during pregnancy will be important in unraveling a potential association between maternal BMI and development of NEC.

## Supporting information

**S1 File. A supporting information file containing de-identified raw data used for analysis has been provided and is available upon request.**
(XLSX)

## Author Contributions

**Conceptualization:** Katherine Stumpf, Priya Sharma, Julie Mirpuri.

**Data curation:** Priya Sharma, Julie Mirpuri.

**Formal analysis:** Katherine Stumpf, L. Steven Brown, Luc P. Brion, Julie Mirpuri.

**Funding acquisition:** Julie Mirpuri.

**Methodology:** Julie Mirpuri.

**Supervision:** Luc P. Brion, Julie Mirpuri.

**Validation:** L. Steven Brown.

**Writing – original draft:** Katherine Stumpf, Priya Sharma.

**Writing – review & editing:** Katherine Stumpf, Luc P. Brion, Julie Mirpuri.

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
