## [Decision Letter · Decision Letter 0]

28 Sep 2023

PONE-D-23-26307Maternal body mass index and necrotizing enterocolitis: A case-control studyPLOS ONE

Dear Dr. Stumpf,

Thank you for submitting your manuscript to PLOS ONE. After careful consideration, we feel that it has merit but does not fully meet PLOS ONE’s publication criteria as it currently stands. Therefore, we invite you to submit a revised version of the manuscript that addresses the points raised during the review process. 1. Introduction:  Please include pathophysiological basis on the association of maternal BMI and NEC2. Method: Kindly include 'lack of gastrointestinal diagnosis' in the section where you describe selection of control group'3. Results: Pleas provide better quality figures4. Conclusion: Kindly change the subheading to discussion and include a separate subheading for the conclusion5. Discussion: Please explain why the large number of Hispanic women is a limitation.7. References: Kindly review reference number 9

We look forward to receiving your revised manuscript.

Kind regards,

Hlengani Lawrence Chauke, MBCHB, BTh, Dip HIV Man, FCOG, MMED (O &G), MSc

Academic Editor

PLOS ONE

Upon re-submitting your revised manuscript, please upload your study’s minimal underlying data set as either Supporting Information files or to a stable, public repository and include the relevant URLs, DOIs, or accession numbers within your revised cover letter. For a list of acceptable repositories, please see http://journals.plos.org/plosone/s/data-availability#loc-recommended-repositories. Any potentially identifying patient information must be fully anonymized

Additional Editor Comments:

Dear Professor, Stumpf

Thank you very much for submitting your manuscript to PLOSONE for consideration. We are grateful that you chose to share your research with our journal. The paper is well written, easy to follow and addresses an interesting topic. The study design and research methodology are aligned with the research questions. The introduction provides a compelling justification for the study. The choice of statistical parameters, data analysis Furthermore the data analysis, data analysis and interpretation are appropriate. The discussion section is scholarly written and demonstrates acquittance with the literature on the subject. The conclusion drawn is also appropriate. I enjoyed reading the paper. The following suggestions could further strengthen the paper:

1. Introduction: Consider including a sentence or two on the pathophysiology behind the association of maternal BMI and NEC.

2. Methodology: -Please add "lack of gastrointestinal diagnoses to the methods describing selection of the control group as recommended by one of the reviewers.

3. Results: Is there a way you can improve the quality of the figures? Please provide an explanation why you consider the number of Hispanic women to be a limitation.

4. Discussion: Please change the subheading 'conclusion to discussion) and also include a comment on the significance of the study's findings.

5. Conclusion: Suggest you add a separate subheading after discussion for the conclusion

6. References: Kindly review reference 9

Congratulations for a well written paper. Looking forward to the revised version.

Kind regards,

Lawrence Chauke

Reviewers' comments:

Reviewer's Responses to Questions

**Comments to the Author**

1. Is the manuscript technically sound, and do the data support the conclusion? 

Reviewer #1: Yes

Reviewer #2: Yes

Reviewer #3: Partly

2. Has the statistical analysis been performed appropriately and rigorously? 

Reviewer #1: Yes

Reviewer #2: Yes

Reviewer #3: Yes

3. Have the authors made all data underlying the findings in their manuscript fully available?

Reviewer #1: Yes

Reviewer #2: Yes

Reviewer #3: Yes

4. Is the manuscript presented in an intelligible fashion and written in standard English?

Reviewer #1: Yes

Reviewer #2: Yes

Reviewer #3: Yes

5. Review Comments to the Author

Reviewer #1: Thank you for the opportunity to review this manuscript. The submission is well written with good research methodology. I have a few minor comments:

-Please add "lack of gastrointestinal diagnosis to the methods describing selection of the control group.

-The sub-heading "Conclusion" should be "Discussion."

-reference 9 is incomplete

-The image quality of the figures is poor

Reviewer #2: The authors have conducted a 10-year retrospective review of NICU admissions at a single urban delivery center to determine if Maternal BMI is associated with Stage 2/3 NEC in preterm babies <33 weeks' gestation. The methodology is a case control study, with ratios of 2 cases: 1 control.

The sample sizes are small but a power analysis was conducted to determine if the sample size available was sufficient to detect any association.

The study is important, given the current levels of obesity in all communities globally and it therefore provides an opportunity to drive the public health messages around adverse pregnancy outcomes associated with high maternal BMI.

The manuscript is well written. The data analysis is sound - and the results appropriately capture what was found in the analysis.

The manuscript however can be improved and/or enhanced by:

Introduction:

The authors need to consider making a strong case for the study by providing an explanation or pathophysiological basis (mechanism) that makes an increased maternal BMI a risk factor for NEC - with a particular focus on the effect of "fat" on placental vasculature and eventually placental insufficiency. I think this will allow even generalists to understand why this study was necessary.

Results and Conclusion

The limitations of the study are well described. It is not clear why the number of Hispanic women in the study was a limitation? Please provide clarity in the manuscript.

Reviewer #3: Dear author

This is an important topic, the results will help identify additional maternal risk factors for NEC.

However, the paper requires revision. There need to be a discussion on the significant findings from your results.

The pathophysiology of NEC and association with BMI should be described in the introduction.

Kind regards

6. PLOS authors have the option to publish the peer review history of their article (what does this mean?). If published, this will include your full peer review and any attached files.

Reviewer #1: **Yes: **Daynia E. Ballot

Reviewer #2: **Yes: **Prof Dini Mawela

Reviewer #3: **Yes: **Dr Tanusha Ramdin

---

## [Author Response · Author response to Decision Letter 0]

7 Dec 2023

All the reviewers’ points have been addressed in our updated manuscript which we summarize below and are highlighted in our manuscript:

1. Editor and reviewers: “Consider including a sentence or two on the pathophysiology behind the association of maternal BMI an NEC”. We have included an additional reference regarding the pathophysiological basis for the association of maternal BMI and necrotizing enterocolitis. This is now addressed in both the introduction and discussion sections.

2. Editor and Reviewer 1: Please add “lack of gastrointestinal diagnoses to the methods”. We have included the reviewers’ suggestion of phrasing to state “lack of a gastrointestinal diagnosis” to describe our control group in the methods section.

3. Editor and reviewers: “Is there a way you can improve on the figures?” We apologize that our figures were not transferred in high resolution as a pdf. Our figures have been updated in the attachments which are clearer and appropriate resolution.

4. Editor: Change subheading. We have changed the subheadings as requested to include both “Discussion” and “Conclusion”. We have included a comment on the significance in our conclusion.

5. Editor and Reviewer: Please provide an explanation on why you consider the number of Hispanic women to be a limitation. An additional explanation regarding limitations of generalizability due to the large number of Hispanic women is now added to the discussion.

6. Editor and Reviewer 1: Kindly review reference 9. Reference 9 has been updated and completed.

Additionally, our data set is fully available upon request and had been uploaded to the UTSW Data Repository.

---

## [Editor Report · Decision Letter 1]

18 Dec 2023

Maternal body mass index and necrotizing enterocolitis: A case-control study

PONE-D-23-26307R1

Dear PROF STUMPF_KATHERINE,

Thank you for the revised manuscript. We’re pleased to inform you that your manuscript has been judged scientifically suitable for publication and will be formally accepted for publication once it meets all outstanding technical requirements.

Kind regards,

 Lawrence Chauke, PhD, MMED (O &G), MSc (Clinical Research), Cert MFM(SA), FCOG(SA), MBCHB, BTh, Dip HIV MAN(SA)

Academic Editor

PLOS ONE

---

## [Editor Report · Acceptance letter]

10 Jan 2024

PONE-D-23-26307R1 

PLOS ONE

Dear Dr. Stumpf, 

I'm pleased to inform you that your manuscript has been deemed suitable for publication in PLOS ONE. Congratulations! Your manuscript is now being handed over to our production team.

Kind regards, 

on behalf of

Prof Hlengani Lawrence Chauke 

Academic Editor

PLOS ONE